# ABLATOR: Robust Horizontal-Scaling of Machine Learning Ablation Experiments

Iordanis Fostiropoulos[1]  Laurent Itti[1]

[1]University of Southern California, Los Angeles California

**Abstract**  Understanding the efficacy of a method requires ablation experiments. Current Machine Learning (ML) workflows emphasize the vertical scaling of large models with paradigms such as 'data-parallelism' or 'model-parallelism'. As a consequence, there is a lack of methods for horizontal scaling of multiple experimental trials. Horizontal scaling is labor intensive when different tools are used for different experiment stages, such as for hyper-parameter optimization, distributed execution, or the consolidation of artifacts. We identify that errors in earlier stages of experimentation propagate to the analysis. Based on our observations, experimental results, and the current literature, we provide recommendations on best practices to prevent errors. To reduce the effort required to perform an accurate analysis and address common errors when scaling the execution of multiple experiments, we introduce **ABLATOR**. Our framework uses a **stateful** experiment design paradigm that provides experiment *persistence* and is *robust* to errors. Our *actionable* analysis artifacts are automatically produced by the *experiment state* and reduce the time to evaluate a hypothesis. We evaluate ABLATOR with ablation studies on a Transformer model, 'Tablator', where we study the effect of 6 architectural components, 8 model hyperparameters, 3 training hyperparameters, and 4 dataset preprocessing methodologies on 11 tabular datasets. We performed the largest ablation experiment for tabular data on Transformer models to date, evaluating 2,337 models in total. Finally, we open source ABLATOR; https://github.com/fostiropoulos/ablator

## 1 Introduction

Machine Learning (ML) research has been criticized for an inability to explain the reasons a method provides an improvement on a specific benchmark. It can be unclear whether a novel component is responsible for the improvement or result of a statistical outlier [35].

Ablation is used to understand how the hyperparameters and architectural components contribute to the performance of a method. This is in contrast to Hyper-Parameter Optimization (HPO) or Neural Architecture Search (NAS) where the objective is to search for the single best performing configuration. As the complexity of ML models increases so does the number of components and parameters that need to be ablated, which increases the search space of possible configurations. Therefore, efficient *horizontal-scaling* of multiple parallel experimental **trials** is necessary.

There are lack of available frameworks for horizontal scaling of ablation experiments. Currently, ML practitioners manually perform horizontal scaling for experiments, such as for hyperparameter selection, distributed execution, consolidation, and analysis of artifacts [10]. Additionally, current frameworks [31] for distributed execution do not provide native support for maintaining the state of an experiment and resuming the execution of multiple trials, referred to as *experiment persistence*. We find that errors in the early stages of experiments can propagate to the analysis and lead to misleading conclusions. Possible errors may be introduced from sampling bias in the hyperparameter selection strategy or the distributed execution fault-intolerance, *survival bias*.

The execution of randomized control trials is necessary to determine causal effects [23, 20]. We identify several sources of errors that can influence the results. We categorize them as Analysis, Execution, and Implementation errors. Analysis errors can result from the hyperparameter selection

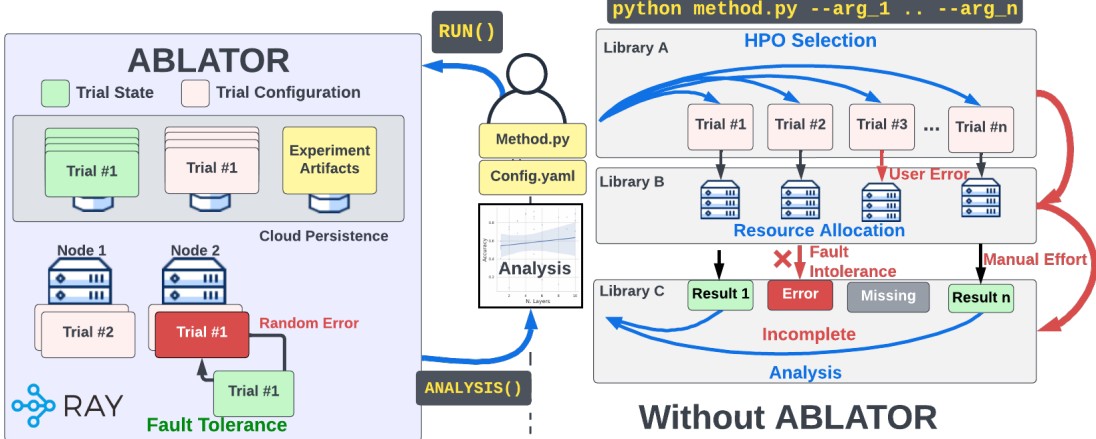

Figure 1: **Left** is the rapid prototyping process when using ABLATOR where only the method implementation and the configuration is required to RUN() the study and provide ANALAYSIS(). ABLATOR handles the horizontal scaling of experimental trials on a cluster of nodes and is fault tolerant, where trials can be continued on the same or different node due to the *Persistence* provided by ABLATOR. **Right** is the process without ABLATOR where the user must use different Libraries or manually perform, 'HPO Selection', 'Resource Allocation', 'Analysis'. Additional *Manual Effort* will be required to integrate between the libraries, where errors between different steps propagate to the analysis that will be erroneous. ABLATOR provides automation by removing boiler-plate code and managing errors internally.

sampling bias. Nonrandom effects during experiment execution can introduce analysis errors. For example, inconclusive trials due to out-of-memory errors caused by a larger model footprint would introduce survival bias to the analysis that will favor smaller models. Implementation errors are mistakes made by users caused by the increased code complexity of ablating multiple method components while maintaining different code bases. We discuss the details of our analysis in Section 3.2.

To aid in error-free horizontal scaling of multiple experiments in ML community, we propose a *stateful* experiment paradigm where we unify all experiment stages under a single framework. A *stateful experiment* is initialized by the *configuration* and code *implementation* of a method. Our framework maintains the state of each experimental trial and provides experiment *persistence*, where the experiment can continue the execution agnostic to the execution environment. The analysis artifacts are produced automatically by the experiment state for faster prototyping. Our paradigm is implemented in our tool **ABLATOR** with support for PyTorch [33] model development. We present an analysis of the sources of errors and provide recommendations that can be useful beyond our framework. We use our framework to study the effect of multiple training and model components on the performance of a Transformer model for tabular dataset 'Tablator' where we perform a large scale ablation study of 2,337 trials. Our contributions can be summarized: **First**; We provide a formalization of a *stateful* experiment design paradigm that we use to address common errors in the execution of ML experiments. **Second**; ABLATOR, a framework that implements our paradigm and facilitate the automated execution and analysis of a model implementation given a configuration. **Third**; We identify sources of error in ML ablation studies and provide recommendations for mitigating them. **Fourth**; We perform the largest to date ablation study of Deep Learning model on Tabular dataset and provide analysis that can be useful to the research community.

We first introduce the features of ABLATOR relevant to horizontal scaling of experiments. Next, we evaluate the main features of our tool in a case study demonstrating the horizontal scaling capabilities of ABLATOR. We present our results using three research questions Sections 3.1 to 3.3.

## 2 Methods

To implement ABLATOR and address common issues in horizontal scaling of experiments, it is necessary to introduce the formalism of a 'stateful experiment design' paradigm. In this section, we introduce our paradigm and in Section 2.4 the implementation of ABLATOR. We identify three stages of an experiment, the design, execution, and analysis (Sections 2.1 to 2.3).

### 2.1 Experiment Design

During the design phase of an ML ablation study, a hypothesis is defined as an *experiment* on the improvement that an architectural component, such as Residual Connections, provides to the performance of the model. The *search-space* of our hypothesis can be defined as Residual = [True, False]. The methodology of our experiment is defined by the *implementation* of the model.

Multiple experimental *trials* are required to improve the statistical power of a test [20] that require *randomly sampling* from the *search-space*. An experimental **trial** can be described as a *stochastic process* that produces a performance *metric*. The *stochasticity* can be observed when performance differs significantly with identical initial conditions, such as re-running the same experiment but obtaining different results.

Thus, to define a *trial*, we maintain two states to describe the system at any given point. The initial conditions (Sections 2.1.1 and 2.1.2) and the current state (Section 2.2). The initial conditions of a trial are defined by the sampled *hyper-parameters* and the *implementation*.

```
distributed.yaml
total_trials: 2000
optim_metrics: [[val_loss, min]]
tune:
  train_config.
    optimizer_config.
      name: ["adam", ....
  train_config.dataset: ["year","yahoo","helena", ...
  model_config.mask_type: ["mix","global","full","random"]
  model_config.residual: [True, False]
  model_config.random_mask_alpha: [0.5, 1]
```

```
prototyping.yaml
train_config:
  dataset: adult
  optimizer_config:
    name: adam
model_config:
  mask_type: random
```

```
1   @configclass
2   class TablatorConfig(ModelConfig):
3       residual: bool = True
4       d_out: Derived[ty.Optional[int]] = None
5       mask_type: MaskType = MaskType("random")
6
7   @configclass
8   class RunConfig(ParallelConfig):
9       experiment_dir: Stateless[Optional[str]] = None
10      model_config: ModelConfig
11      train_config: TrainConfig
```

Figure 2: ABLATOR provides a configuration system specific to ML experiments, where it has to encompass multiple trials in a compact definition and be unambiguous. On **left**, is an *illustration* of the configuration for distributed execution (**distributed.yaml**) and method prototyping (**prototyping.yaml**). On the **right**, the configuration is type checked by the ABLATOR library. The library provides flexible type definitions (red) that are resolved during run-time. The configuration is compact and unambiguous at initialization, supporting our stateful experiment design paradigm in Section 2.1.

### 2.1.1 Configuration

describes the hyperparameter **search-space** from which the hyperparameters are sampled. Two custom Python annotations are introduced, Stateless and Derived, to define attributes to which the experiment state is agnostic, while unannotated attributes are assumed to be **stateful** control variables. Stateful attributes require an assignment during the initialization stage unless they are annotated as Optional.

**Stateless** configuration attributes can be used as a proxy for variables that can take different value assignments between trials or experiments. For example, the learning rate can be set as an independent variable and *must* be annotated as stateless. Additionally, there are variables that take different values between experiments and trials to which the state is agnostic, for example, a random seed or a directory path between execution environments *can* be annotated as stateless.

**Derived** attributes are un-decided at the start of the experiment and do not require a value assignment. Instead, the value is determined by internal experiment processes that can depend on other experimental attributes, such as the dataset. However, given the same initial state, the attribute is expected to result in the same value and is therefore *deterministic*. For example, the

input size used in a model's architecture that depends on the dataset will be annotated as `Derived` during the experiment design phase.

The annotations address common requirements of ML experiments, where a configuration may have to describe a search-space that encompasses multiple trials, as opposed to taking on a specific value assignment at initialization. Additionally, an ML experiment can have attributes that are difficult to model at initialization but can be inferred during execution. For a stateful design paradigm, the configuration should be unambiguous at the initialization state, i.e. Figure 2.

**2.1.2 Implementation.** The implementation describes the methodology of the hypothesis. **Invariance** of the implementation w.r.t. the method evaluated produces a single code artifact that encapsulates all methods i.e. a single code base for using and not using residual connections. The implementation computes one or more evaluation metrics. Lastly, the implementation should have a deterministic value assignment to the variables we defined as `Derived`.

Implementation invariance provides a compact representation and is robust to errors. A compact representation provides *ease of use* that is a consequence of a shared implementation among the ablating components where the differences are specified through the configuration and applied by conditional `if` statements. The advantage of this approach is that the performance variance caused by implementation differences is minimized, where even the order of matrix multiplication can have significant effects on the method performance [46].

## 2.2 Experiment Execution

**Experiment state** can be `Running` or `Complete` as the aggregate of the state of all experimental *trials*. Each **trial** can be in three additional states as `Pending`, `Failed` or `Pruned`. **Pending** trials are defined by their initial conditions alone, i.e. the sampled hyperparameters. A **Running** trial extends the definition to include a *checkpoint*. **Complete** trials extends the definition to include one or more *metrics*, such as the validation loss. **Pruned** and **Failed** trials are a result of irrecoverable errors during initialization or execution. A **fault-tolerant** strategy reschedules trials with recoverable errors as *Pending* and attempts to resume from the *checkpoint*. A long-running experiment can be interrupted (i.e. server maintenance) while errored trials do not interfere with the results (i.e. failed trials due to recoverable errors).

**Checkpoint** describes the optimization state of a trial and contains sufficient information to resume execution. `ABLATOR` store the model weights, optimizer, scheduler, and training meta-data such as current training iteration using a compact representation. The checkpoint mechanism in `ABLATOR` can be extended to support custom use cases, i.e. RL. Lastly, maintaining the state of the experiment requires keeping track of the checkpoints and results. Multiple checkpoints are stored locally on each node and can be synchronized with cloud storage. The experiment is agnostic to the execution environment; **experiment persistence**.

## 2.3 Actionable Analysis

Analysis that is **actionable**, is a result of the automation to provide sufficient artifacts to support decision making. The artifacts should help facilitate a quick and informed decision on the likelihood of the hypothesis. The experiment state is used to infer the hypothesis, i.e. 'what are we ablating?', and conclusiveness of the analysis i.e. 'is the trial failed?'. The analyses `ABLATOR` provides infer the search-space, such as control and independent variables from the configuration and the variable `type` to produce the corresponding artifacts. The artifacts produced address common problems in evaluating ML methods (Section 3.2). For each attribute, the goal is to encapsulate the best, average, variance and distribution of the performance metric under a single figure; i.e. Figures 4 and 5.

## 2.4 ABLATOR

`ABLATOR` is designed in Python and with support for PyTorch models, while the distributed execution system uses Ray Core [31]; Figure 1. We describe the features of `ABLATOR` important in addressing

a stateful experiment paradigm. `ABLATOR` can be extended or customized specific to the use-case without loss of automation where an object-oriented design provide access to function overwriting. The features of `ABLATOR` provide **ease of use** where it requires defining an experiment through implementation and configuration. **Automation** is supported by providing an abstraction layer on distributed execution with fault tolerance, artifact consolidation, and analysis. Our framework is agnostic to the execution environment and can run on a laptop and a cluster of nodes.

**Configuration** use a hierarchical dictionary-like format that is easy to understand and can be converted to and from *yaml* files. `ABLATOR` uses a strict type-checking system with custom annotations (Section 2.1.1). A unique signature identifier ("ID") is generated for each experiment that corresponds to the values of the stateful configuration attributes, while for a trial, the identifier is based on the unique value assignment of all configurable properties. Thus, the configuration system allows for a hierarchical representation of trials under a single experiment and facilitate experiment persistence where multiple experiments are stored in the same directory.

**Implementation** A `Trainer` class will manage the physical resources of the experiment. There are two options according to the use case, `ProtoTrainer` for prototyping at a local environment, and `ParallelTrainer` for horizontal scaling of a single experiment. `ParallelTrainer` is unique to `ABLATOR`, where multiple trials are managed and executed in parallel. Prototyping to experiment deployment requires a single change `ProtoTrainer` $\implies$ `ParallelTrainer`.

**Artifact Persistence** For every resource node, the trials are executed in parallel, and failure in a single trial does not result in interruption of the experiment. We use the master node to maintain the experiment state (Section 2.2) and synchronize the artifacts of all nodes with a central database. Cloud compute nodes are often ephemeral, and restarting the experiment requires only for the files to be synchronized among the centralized storage and all nodes. Furthermore, the files stored in the central storage are sufficient to perform an analysis or recover from errors.

**Analysis Artifacts** are specific to numerical attributes and categorical attributes. The attribute type is informed by the configuration. `Figure` are artifacts that summarize the mean, best, and distribution of a performance metric. For numerical attributes, we use scatter-plot with optional interpolation curves while for categorical attributes we use violin-plots. The analysis can be extended to support custom use cases, such as additional figures or tables, while still being automatically generated from the experiment state; examples are in Section 3.3 and our supplementary.

## 3 Experiments and Results

We first present how `ABLATOR` can be used for horizontal scaling with an ablation study on the 'Tablator', a Transformer model we designed for this study; Section 3.1. In Section 3.2 we categorize common errors during horizontal scaling of ablation experiments and provide our recommendations. In Section 3.3 we provide the results of an ablation experiment on tabular dataset benchmark. For reasons of brevity, we discuss only the results most relevant to `ABLATOR`. We attach the code that was used for our experiments and analysis, and additional experiments in the supplementary.

### 3.1 RQ-1: How can `ABLATOR` improve the horizontal scaling of thousand experimental trials?

`ABLATOR` requires the configuration and implementation. We extend the implementation of FT-Transformers (FT-T) [1] [17] with minimal changes to the original code. We implement a model we call 'Tablator' and evaluate all the design components of FT-T as well as the effect of Residual Connections [21] and Attention Masks inspired by BigBird [45]. We evaluate 'Full', 'Mixed', 'Global', and 'Random' attention mechanisms and explain their implementation in the supplementary.

We perform an ablation on 14 model hyperparameters and components in total, and evaluate the effect model-capacity, dropout hyper-parameters , prenormalization, weight initialization, and activation function have on the model performance. Additionally, we evaluate 7 dataset

---

[1] https://github.com/Yura52/tabular-dl-revisiting-models

preprocessing techniques and training configurations, such as feature encoding methods, missing value imputation, feature normalization, training time, optimization.

The differences between 'Tablator' and FT-T are on an additional module for Attention masks that requires 9 additional lines of code as well as 2 lines of code insertions for residual connections. The majority of the development effort was directed towards making the original dataset performant and converting it to a PyTorch `Dataset` as opposed to a Python `dataclass`. We define the tunable configurable hyperparameters as shown in Figure 2.

We first verified our implementation with a `ProtoTrainer` in this section and then we scale our experiment with a single code change using a `ParallelTrainer` to thousands of trials for our results in Section 3.3. For this experiment, it took significantly more time to write the current section of this paper than it took to write the code and start the execution of the experiments.

### 3.2 RQ-2: What are common sources of errors during horizontal scaling of experiments?

We identify 3 categories of errors Analysis †, Execution ‡ and Implemention∗ errors that are based on empirical observations and use previous analysis [10, 8, 9, 27, 36, 1, 46, 12] to support our conclusions. In this section, we provide examples of each and attach additional analysis in our supplementary.

**Sampling Strategy** † can be incompatible with the method used to evaluate the performance of a component and lead to misleading analysis [41]. For example, performing HPO and comparing the mean performance of the sampled trials can bias the result towards a single component variant. We perform two identical experiments using Tablator with an identical budget for CovType ('CO') dataset [7]. When random sampling between 5 optimizers AdaB [47], Adam[24], AdamW [29], RAdam[28], SGD[39] every optimization algorithm was sampled with an even probability $P(\mathcal{O}) \approx 0.2$. Contrary, when performing HPO with Tree-structured Parzen Estimator (TPE) [3], SGD was oversampled with $P(SGD) = 0.76$ as it was found to perform relatively better compared to other methods. Other optimization methods were undersampled by TPE and their estimated performance is lower when compared to the empirical mean performance of the same method calculated via Random Sampling. When TPE was used, all optimizers *appeared* to underperform on average by 4.6% and 3.8% when evaluating the best and mean trial performance. We conclude that statistical tests can be influenced by the bias of the HPO method used to sample configurations and their performance might not be fully explored.

**Survival Bias** † can be caused by nonrandom execution errors. We identify the trials for which there were memory errors. We perform feature importance analysis and use a surrogate random forest model [34] to predict whether a trial will result in a memory error. We find that the configuration attributes related to the dataset and the hidden di-

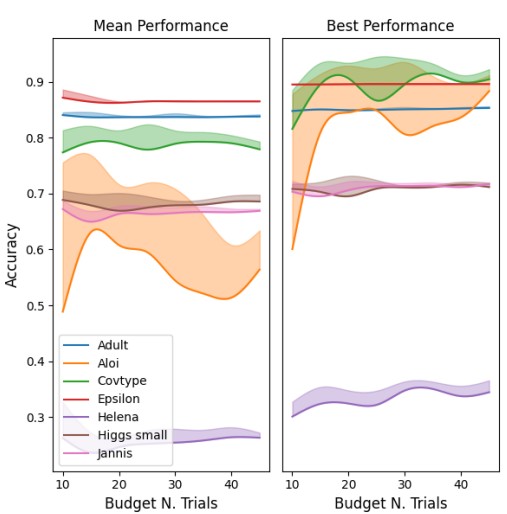

Figure 3: We evaluate how **Budget Allocation**‡ can influence the analysis of an ablation study. We vary the number of trials we use for analysis ('*N* trials'). We compare estimating the performance of a method to a dataset using the mean (**left**) (i.e. ANOVA) or the best (**right**) trial (i.e. proof-by-existence). Evaluating the performance of a component by its mean performance would require fewer trials for easier dataset ('Covtype') when compared to using the best trial. While for more challenging dataset ('Aloi') evaluating by the best trial would be more efficient, as the performance converges at around 20 trials (right figure) compared to >50 for the mean (left figure). We conclude that the ablation budget should be taken into account and relevant to the type of analysis.

| Dataset | CA ↓ | AD ↑ | HE ↑ | JA ↑ | HI ↑ | AL ↑ | EP ↑ | YE ↓ | CO ↑ | YA ↓ | MI ↓ |
|---|---|---|---|---|---|---|---|---|---|---|---|
| FT-T | 0.459 | 0.859 | 0.391 | 0.732 | 0.729 | 0.960 | 0.898 | 8.855 | 0.970 | 0.756 | 0.746 |
| Tablator | 0.535 | 0.856 | 0.368 | 0.718 | 0.723 | 0.921 | 0.896 | 8.778 | 0.930 | 0.780 | 0.749 |
| $\Delta Imp.*$ | -0.076 | 0.003 | 0.023 | 0.014 | 0.006 | 0.039 | 0.002 | 0.077 | 0.04 | -0.024 | -0.003 |

Table 1: We evaluate the difference between the best performing trials as reported by FT-Transformer ('FT-T')[17] and as found by our ablation experiments in Section 2.1. FT-T is in the subspace of configurations of Tablator where a greedy HPO strategy is used as opposed to random sampling for Tablator. As such, we expect Tablator to perform similarly but **not** better. We use the benchmark as a way to evaluate Implementation Errors ∗ from Section 3.2. We conclude that our implementation contains no errors, as the relative difference ($\Delta Imp.*$) is within the expected margin of error between HPO and random sampling.

mension were the most important. A larger dataset has more features, which leads to a model with larger hidden dimension. The attributes related to the hidden dimension scored 23% higher than the average feature importance. We conclude that smaller models and dataset will have a Survival Bias from the fewer out-of-memory execution errors and that such bias could be mitigated by better resource allocation. For example, one can group experiments by their memory utilization as to avoid out-of-memory errors from the largest trial.

**Resource Utilization statistics** ‡ We observe the resource utilization statistics, the mean usage of a trial is 3,075 ± 3,578 (MiB) while the maximum is 32,303 (MiB). The high variance in memory utilization is a consequence of a search space that correlates with memory utilization. Allocating resources based on the largest trial might be infeasible. Using a heuristic for resource utilization might be necessary.

**Budget Allocation** ‡ we vary the number of experimental trials for 10 repeated observations and report the best and mean performance in Figure 3. An increased budget reduces the variance of the mean performance. We report less variance in the performance of the best trial for repeated observations. We conclude that, for 'Tablator', fewer trials are required to obtain an estimate of the top performance while the mean performance would require more trials.

**Implementation Errors** ∗ Our observations on implementation errors extend previous analysis [46, 27, 36, 12] on the impact of ML tooling where the sources of errors are poor development practices and variance introduced by tooling. Packaging has the benefit of incremental development and modular design, where in the example of 'Tablator' two methods ([45] and [17]) can be combined. Additionally, as the method complexity increases, version control that includes the configuration, and analysis that corresponds to the implementation can prevent misinterpretation of the results.

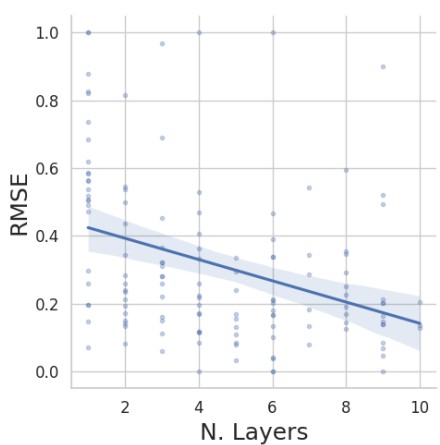

Figure 4: Evaluation of the effect of a larger model for a regression data set, where (RMSE) ↓ is normalized for the relative difficulty of each dataset. Larger model performs better but with higher variance where the uncertainty on the estimated performance increases. A larger model might be a more risky choice when deploying a model that requires to be iteratively trained.

## 3.3 RQ-3: Can `ABLATOR` be used to perform a large-scale ablation study on Tabular Dataset?

We use 'Tablator' presented in Section 3.1 to evaluate possible improvements in data processing, the Transformer model architecture, and the effect of training hyperparameters on 2,337 trials,

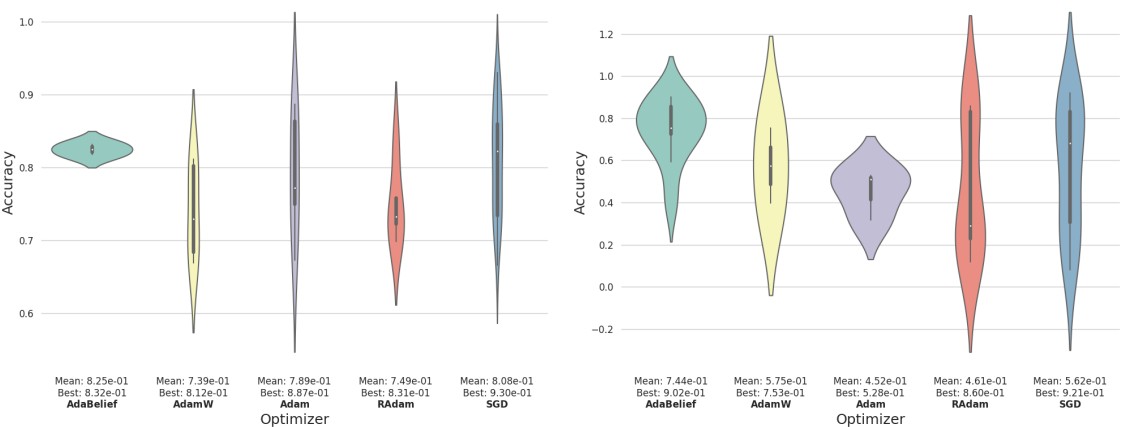

Figure 5: Example of Automatically generated analysis artifacts from ABLATOR. On the **left** are the artifacts for 'CO' [7] and on the **right** for 'AL' [16]. We compare the effect of an Optimizer on the performance to a dataset. In agreement with [44], there is no single model that generalizes across all dataset; where for example Adam [24] under-performs for 'AL' but not for 'CO'. We conclude that seperate ablation studies will be required for different dataset.

where the current largest ablation on tabular dataset is 2,000 trials [48]. Our results are summarized in Figures 4 and 5. On Table 1 we report the Accuracy, where higher is better ↑ and root square-mean-error ('RMSE') where lower is better ↓ on 11 dataset; [32, 25, 18, 18, 2, 16, 17, 4, 7, 11, 38] identical to the benchmark of FT-T [17]. We find Tablator performs similarly in all datasets. The goal of the benchmark comparison is to verify our implementation, while the goal of our study is to evaluate general methods that work best among dataset and not a benchmark improvement. Similarly to FT-T [17], we conclude that the simplest methods work best in most general cases, i.e. SGD [39] with momentum has the best mean performance on 9 of 11 datasets. For more complex methods, there is a large variance on the performance of the method between datasets.

For example, we find that RAdam [28] ranks on average 2.71 for classification dataset but 3.75 for regression dataset when evaluated by the mean performance. Additionally, more complex methods may result in the best performing trial but perform worse on average, where RAdam ranks on average 2.25 when evaluated on the best-performing trial for regression dataset (compared to 3.75). Our results indicate that using a complex method may require a large tuning budget to return good results. Additionally, we conclude that larger models only perform moderately better Figure 4.

The high-performance variance between different components on different datasets leads us to conclude that evaluations should be done with multiple datasets. Additionally, we find that tuning would be required that is specific to the dataset and the training configuration. Simple design choices, such as SGD and moderate model capacity, can provide a good starting point, while more complex training configurations can provide trade-offs on performance and uncertainty that can be specific to the use case.

From the median and mean performance observed in our results, we did not find that any of the preprocessing methods to have a consistent, significant effect on the model performance. ABLATOR can help provide actionable results specific to the dataset. We conclude that several ablation experiments are required to evaluate a method and ABLATOR is the only tool currently available to facilitate rapid evaluation.

## 4 Discussion

In our work we present ABLATOR an AutoML framework for ablation experiments. Beyond our framework, there are several issues w.r.t. automated decision making as there is no universal

statistical test or threshold to accept or reject a hypothesis. Analysis requires domain expertise relevant to the evaluation setting. Specific to ML research is the lack of methods for evaluation of a hypothesis where the metric can be both non-normally distributed and heteroskedastic i.e. Figure 5.

**Broader Impact Statement** Performing large-scale ablation experiments may require a large number of computational resources that can negatively impact the environment through $CO_2$ emissions. However, the automation provided by ABLATOR can result in a more effective use of computational resources and reduce $CO_2$ emissions. ABLATOR can help improve research practices without a negative impact on society when used in the context in which it is presented.

## 5 Related Works

We identify four categories of work that are most similar to ours. Work that focuses on errors introduced by tools and incorrect analysis, on horizontal scaling of experiments, works that aid in ablation studies, and tools for automated HPO.

Previous work [10, 8, 9, 27, 36, 1, 46, 12] identify the **source of erroneous analysis** as poor experiment design practices resulting from improper use of statistical evaluation methods, HPO budget, HPO strategies, and tooling and provide recommendations. We extend their work and investigate errors during horizontal scaling of experiments that lead to erroneous analysis. We identify errors from the sampling strategy, non-random execution errors, and implementation errors. We provide general recommendations in Section 3.2 and address the errors with ABLATOR.

Several tools are proposed [13, 15, 22, 43, 26] that support **distributed experiment execution**. However, they require manual effort in integrating with other libraries for resource allocation, scheduling of experiments, resuming faulty trials, result aggregation, configuration sampling, and analysis. Contrary, ABLATOR combine all of the above in an automated fashion, where only the implementation and configuration of the method are used to produce the analysis artifacts.

**Ablation** framework introduce methods and tools specific to constructing ablation analysis artifacts. Such methods can have limited use cases [19, 5, 37] or lack automation [42]. In contrast, ABLATOR provides analysis artifacts that provide a holistic view of a method's performance that can be extended to support automation and specific use-cases addressed by the works above.

**AutoML** methods [14, 48, 6] are designed for HPO and can be extended to ablation experiments that provide support for automated analysis. Unlike ABLATOR, such tools are designed for simple use cases, such as statistical models, and require additional effort to scale the experiments horizontally. Such tools and similar, can be used as the implementation provided to ABLATOR and as such are orthogonal to our work. AutoAblation [40] extends Maggy [30] to Deep Learning models. However, allocating and managing GPU resources for each trial requires manual effort. While AutoAblation does not provide experiment persistence and as such is not fault-tolerant. Additionally, the declarative design paradigm has limited use cases, as opposed to the object-oriented design of ABLATOR.

As such, ABLATOR improves automation by managing GPU resources, storing of experimental artifacts, restarting erroneous trials, removing boiler-plate code where only the method implementation with the configuration is required to provide automated analysis.

## 6 Conclusion

In this work, we identify several sources of error common in horizontal scaling of multiple experimental trials. We provide general recommendations and address errors with a stateful experiment design paradigm. ABLATOR implement the paradigm to automate the scaling of ablation experiments across multiple resources and produce analysis artifacts in an automated fashion and for rapid iterative prototyping. We evaluate ABLATOR with a Transformer model for Tabular dataset, 'Tablator', where we study the effect of several architectural components and hyperparameters on the largest ablation study for tabular dataset to-date. ABLATOR is an effect tool to conduct large-scale ablation studies with ease and lead to actionable insights that are particular to the experimental setting.

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
