# OpenReview forum: "ABLATOR: Robust Horizontal-Scaling of Machine Learning Ablation Experiments"
_automl.cc/AutoML/2023/ABCD_Track — AutoML 2023 (ABCD Track)_

### Official Review · Reviewer_gPYR · 2023-05-01

**Potential Impact On The Field Of Automl Rating:** 3
**Technical Quality And Correctness Rating:** 3
**Clarity Rating:** 3
**Actions Required To Increase Overall Recommendation:** Address the cons described in overall…

**Summary Of Contributions:**

The paper proposes Abalator, an open source framework for running and managing ablation experiments in machine learning. The framework requires two inputs, the experiment design config which describes the set of ablations and the model implementation. It outputs artifacts which are figures, visualizations comparing the performance of different parameters.
The framework is fault tolerant and persists experiment state to recover from failures. It supports PyTorch models and uses the ray library for distributed execution. The authors evaluate the system using a large ablation study involving 2300 runs.


**Clarity:**

The paper is well written. The motivation, relevant background and methodology are well described.

**Overall Review:**

Pros:

1. Addresses a practical and common problem in ML of running ablation experiments. The framework is built on top of existing tooling which makes it easily extensible.
2. The features for fault tolerance and artifact results generation are useful.
3. The discussion around common source of errors in ablations is insightful,


Cons:
1. Comparison with existing tools (like AutoAblation) and how Ablator is better is limited and needs more detail.
2. The description of how the ablation runs are created from config is missing. Similarly the set of ablation policies and configuration features for setting up the trial aren't discussed in detail.
3. Missing description of limitations.
4. The github page mentions Ablator can be compared with hydra for configuration and ray for HPO, but its unclear how similar or different they are and more documentation is required.


**Potential Impact On The Field Of Automl:**

The framework is aimed at addressing a very common use-case of running and managing ablations. Most of these are run manually, or use variations of HPO frameworks which may not be perfectly suitable. So this framework could be useful.


**Reproducibility (Optional):**

The code is provided

**Review Confidence:**

4: You are confident in your assessment, but not absolutely certain. It is unlikely, but not impossible, that you did not understand some parts of the submission or that you are unfamiliar with some pieces of related work.

**Review Rating:**

7: Weak Accept: Technically sound paper with moderate-to-high impact, with perhaps some minor flaws.

**Review Summary:**

The paper is well written, the proposed framework is useful and relevant for running ablations. More discussion around features and comparison with existing tools is required.

**Technical Quality And Correctness:**

The framework supports most of the features for running ablation experiments and uses existing frameworks wherever relevant like pytorch for training and ray for distributed execution. The support for fault tolerant experiments is useful and the output artifacts comparing the experiments make analysis easy. The description of common sources of errors in ablations are insightful. However comparison with existing tools is limited and it's unclear what ablation policies are available and how flexible the config and framework is.

---

### Official Review · Reviewer_V97J · 2023-05-08

**Potential Impact On The Field Of Automl Rating:** 3
**Technical Quality And Correctness Rating:** 3
**Clarity Rating:** 3

**Summary Of Contributions:**

The paper introduces ABLATOR, a novel framework that facilitates conducting horizontal-scale ablation studies with ease. The framework is implemented based on the "stateful experiment design" paradigm, which is also introduced in this work. This new paradigm encapsulates the design, execution, and analysis of experiments, aiming to provide robustness against errors and facilitate hypothesis evaluation.

**Actions Required To Increase Overall Recommendation:**

All of my concerns regarding the paper have been mentioned above. However, here is a summary of the suggested actions that could further enhance the quality of the work: (1) The versatility of ABLATOR should be demonstrated by performing ablation studies on more models with different data types and addressing various problems. (2) Providing a detailed documentation on how to use and extend ABLATOR, including examples in the GitHub repository, would be highly beneficial. (3) It would be beneficial to include a discussion on the compatibility of ABLATOR with existing libraries and frameworks.

[Update] The authors have addressed all my concerns during the rebuttal. Therefore, I have updated my rating accordingly.

**Clarity:**

Overall, the paper is well-written and provides a detailed description of the experiments. However, some minor changes could enhance its clarity:

* The "Experiment Execution" section could benefit from a more detailed explanation of the "compact representation" used to store model weights, optimizer, scheduler, and training meta-data.
* Including more examples of how to use ABLATOR would be helpful, as there is currently no detailed documentation on how to modify existing code to include ABLATOR.
* While the paper mentions that ABLATOR supports PyTorch, it would be useful to clarify whether other libraries, such as Tensorflow, are also supported.
* The authors mention that ABLATOR can be customized for specific use-cases, but they do not provide guidelines or information on how to do so.
* Section 3.3 should reference Section 3.1 instead of 3.2 for the presentation of Tablator, as Section 3.2 discusses how ABLATOR addresses common sources of errors during horizontal scaling.
* To improve readability, Table 2 could use bold values to indicate the best performance on each dataset.
* Inconsistent referencing of figures is present, with Section 2.4 using "Figure X" and the rest of the paper using "fig. X."

[Update] The authors have updated the paper taking into account my suggestions.

**Overall Review:**

Pros:
* The framework has the potential to simplify horizontal scaling of machine learning ablation experiments.
* The paper identifies common errors that can occur when horizontally scaling ablation studies and provides suggestions to avoid them.
* The framework contains enough documentation and examples on how to use it.

Cons:
* The paper does not provide a broader discussion of ABLATOR's limitations, such as whether it can be used with models of any kind (e.g. classification, regression, segmentation).

**Potential Impact On The Field Of Automl:**

I believe that the impact of ABLATOR is more relevant to the broader artificial intelligence community than just the AutoML community. The primary advantage of this framework is its ability to facilitate horizontal-scale ablation studies. While it may not be the best option for finding the optimal hyperparameters for a model, it is well-suited for analyzing the impact of architectural components or, in general, ablating specific components.

**Reproducibility (Optional):**

The framework is open-source on GitHub, and the authors have made available all the necessary files and information, including hardware resources and computational time, to reproduce the results presented in the paper.

**Review Confidence:**

4: You are confident in your assessment, but not absolutely certain. It is unlikely, but not impossible, that you did not understand some parts of the submission or that you are unfamiliar with some pieces of related work.

**Review Rating:**

7: Weak Accept: Technically sound paper with moderate-to-high impact, with perhaps some minor flaws.

**Review Summary:**

The paper introduces ABLATOR, a novel framework designed to simplify the horizontal scaling of ablation studies. Additionally, the authors identify common errors found in horizontally scaling ablation experiments and provide suggestions to mitigate them, introducing the paradigm of "stateful experiment design" to ensure the robustness of the framework. Although the authors conducted the largest ablation study for tabular data, it would be beneficial to include experiments with other models tackling various problems and data types to showcase the versatility of ABLATOR. A more comprehensive documentation on how to use ABLATOR and how to customize it would also enhance its impact.

[Update] The authors have included a comprehensive documentation on how to use ABLATOR.

**Technical Quality And Correctness:**

Although the paper provides enough information about the benefits of the framework, the authors focused their experiments solely on a single model (Tablator, a transformer model for tabular data) and conducted the largest ablation experiment for tabular data (2337 versions of Tablator). While this demonstrates the potential of ABLATOR, it would be even more interesting to see the framework applied to several models with different data types (e.g., images, graphs, audio) and tackling different problems (e.g., classification, regression, segmentation). This would help to demonstrate the broader capabilities and limitations of ABLATOR.

---

### Official Review · Reviewer_e6Zj · 2023-05-09

**Potential Impact On The Field Of Automl Rating:** 3
**Technical Quality And Correctness Rating:** 3
**Clarity Rating:** 3

**Summary Of Contributions:**

This paper proposes a new software framework for large-scale ablation studies in machine learning pipelines, pytorch models in particular.


**Actions Required To Increase Overall Recommendation:**

The main point I would like to see improved is the clarity of section 3, with regard to more directly separating the particularities of the use-case from the general functionality of the tool.

**Clarity:**

The paper is clearly written, with a formal disucussion about the requirements for performing ablation studies in an ML context being helpful to orient the reader.

The structure of section 3 is somewhat confusing in places. In particular for section 3.2, it is said that there are 3 categories of error, but there are then 5 highlighted subsections, which makes the section structure less intuitive.
It is also somewhat unclear what the budget allocation experiment shows explicitly. Is this meant only as an additional experiment on the selected example, or is this part of ablator itself? If so, are the conclusions drawn here general for the tool, or for this specific usecase? Having a clearer separation between the usecase and the general statements about ablator would help in the interpretation.

Two small notes:
- in 3.3, it is said tablator is introduced in 3.2, while it seems this should be 3.1.
- in secton 2 at the end of the first paragraph: remove 'such as'

**Overall Review:**

The paper provides an interesting perspective on ablation studies in a part of machine learning, and illustrate how their proposed software can help make this type of analysis easier to execute.

The provided example use-case shows hints of its applicability, but it is somewhat limited in scope. The mixing between usecase-specific and general tool-based functionality can be confusing in places, so a clearer separation might help both show the condiderations made in the design as well as the resulting types of data for supporting the user in their analysis.


**Potential Impact On The Field Of Automl:**

Ablation studies can be an important part of many research directions within ML, so providing access to a robust tool to automate part of this process can definitely have an impact. The paper clearly discusses practical considerations in ablation studies, and how these have been made handled in the proposed software package.


**Review Confidence:**

2: You are willing to defend your assessment, but it is quite likely that you did not understand the central parts of the submission or that you are unfamiliar with some pieces of related work.

**Review Rating:**

7: Weak Accept: Technically sound paper with moderate-to-high impact, with perhaps some minor flaws.

**Review Summary:**

The paper clearly describes the challenges with executing ablation studies in the context of large pytorch-models, and proposes a software-based solution which addresses these challenges. While the clarity of the use-case can be improved, the overall paper seems to be relevant to the track.

**Technical Quality And Correctness:**

The technical aspects of the software are discussed clearly, altough the experiment setup in section 3.3 could have used a bit more elaboration.

---

### Official Review · Reviewer_u1c8 · 2023-05-09

**Potential Impact On The Field Of Automl Rating:** 3
**Technical Quality And Correctness Rating:** 2
**Clarity Rating:** 2

**Summary Of Contributions:**

This paper introduces a stateful formulation of horizontal scaling for ablation experiments to combat common errors during development. It instantiates this formulation into the ABLATOR framework in PyTorch.

In addition to the framework, the paper gives recommendations for good practice and performs a big benchmark on tabular data.

**Actions Required To Increase Overall Recommendation:**

My main concerns are:
* The focus of the paper is a bit unclear. I think it would benefit from fully focusing on the ABLATOR framework and showcasing its functionality and effectiveness. This might just involve clarifying better in the text how everything fits together or it might mean exploring questions like how ABLATOR performs compared to alternative approaches in terms of usability and time complexity.
* It would help to have more visual depictions of the framework's functionality and how it compares to other alternatives, both in the paper, code repository and documentation.
* The framework lacks guidance on how to use it, in the form of tutorials or examples. The documentation is very bare-bones and seems to be purely based on automatically generated information from code. Adding these in would massively increase the potential usefulness of this application.

**Clarity:**

The presentation of the framework and evaluation is clear and detailed. However, the focus of the paper seems to be somewhere in between two different things. The first is introducing and benchmarking the proposed package, and the second is doing experimental evaluation to provide insights to the community on ablation scaling. I feel that this split in focus is a bit confusing as neither is fully explored.

**Overall Review:**

### Strengths
* The framework formulation is rigorous and well-thought-through.
* It is an open-source software package released under the GPL 3 license.
* The code repository is very recent but has some following already.
* The package gives broad functionality relevant to the AutoML community.

### Weaknesses
* The codebase and documentation lack tutorials or introductory information.
* It is hard to understand what functionality this framework provides. It would help to have more visual depictions of this, both in the paper, code repository and documentation.
* The codebase has only one contributor, and few issues have been closed.
* The focus of the paper, especially the experimental section, is split between presenting the ABLATOR framework and attempting to provide novel insights. It leads to neither goal being fully achieved.

**Potential Impact On The Field Of Automl:**

Ablation studies and HPO are important aspects of AutoML. This framework has the potential to be useful for practitioners in the field. However, it would first need to be presented in a much clearer manner, comparing it more to existing alternatives and giving much more guidance on its usage.

**Review Confidence:**

3: You are fairly confident in your assessment. It is possible that you did not understand some parts of the submission or that you are unfamiliar with some pieces of related work.

**Review Rating:**

5: Borderline Leaning Reject: Technically sound submission where reasons to reject nonetheless outweigh reasons to accept. Please use sparingly.

**Review Summary:**

A good framework that suffers from poor documentation and a paper that lacks focus and tries to cover too many bases.

**Technical Quality And Correctness:**

The quality of the application is good, with a stateful design at its core. The code seems of good quality and its associated documentation has strong coverage in terms of classes and methods. But the documentation is missing all guidance on how to use it. This is a big downside.

---

### Official Review · Reviewer_dcs4 · 2023-05-11

**Potential Impact On The Field Of Automl Rating:** 3
**Technical Quality And Correctness Rating:** 2
**Clarity Rating:** 3

**Summary Of Contributions:**

The authors propose ABLATOR, a system to horizontally scale ML experiments while reducing the rate of implementation errors and false conclusions via stateful experiment design focused on persistence and robustness. ABLATOR provides actionable analytics automatically from experimental runs, and supports multi-node trial runs using Ray Core as a backend. The authors additionally use ABLATOR to evaluate 'Tabulator', a tabular transformer model, against FT-Transformer and analyze the impact of hyperparameters on the performance across 11 tabular datasets.

**Actions Required To Increase Overall Recommendation:**

I would consider raising my score assuming my concerns are addressed in the rebuttal, particularly regarding Table 1.

[Update 2]: This has been mostly addressed, and thus I updated my score

**Clarity:**

In general the paper is clear and well written, although the nuance of how hyperparameters are searched can be difficult to understand on a first read, as well as how ABLATOR directly tries to resolve each of the different sources of experimental error.

What is the point of evaluating the mean performance of a method compared to the best performance of the method?

Currently you are defining the uncertainty within the context of 1 dataset, but probably the more relevant uncertainty should be computed across all datasets via some form of aggregation to inform the decision of whether "A is better than B on average across N datasets". By leveraging all datasets in the benchmark, this would strengthen your statistical certainty, although I don't know exactly how the aggregation should best be done.

Table 1: Add bolding for best result on a given dataset. Include average rank as additional column.

I'm a bit unclear on how ABLATOR helps prevent implementation errors. It doesn't provide functionality to automatically detect errors, and while it is fair to say that versioning and packaging of experiments is a good practice, it would require that the user have a correct implementation in the first place, and the definition of "correct" in ML is generally not well specified. How does ABLATOR help prevent implementation errors compared to something like AutoMLBenchmark, which uses a benchmark specific test scoring mechanism to ensure scores are being consistently computed for all frameworks and datasets?

It is fairly unclear how you are sampling hyperparameters without very careful reading of the paper. I initially thought you were sampling only 1 hyperparameter at a time with all others fixed, but it appears you are searching multiple hyperparameters at once. A description of the sampling strategy might be good to highlight in a more visible way.

**Overall Review:**

This is a strong paper that proposes a useful experiment execution, tracking, and analysis suite along with relevant code for reproducibility and analysis of insights.

My biggest concern is the strange take-aways from Table 1 that seem to contradict the data, which I would like clarification on by the authors in the rebuttal. I am also a bit confused on the positioning of 'Tablator' in the paper. Is this supposed to be a novel model introduction?

Additionally, it is unclear given all these experiments what the actionable take-away is, as it appears that even with over 2000 trials on 11 datasets that no statistically significant result was observed. What part of this process is "Auto" if anything in regards to "actionable" take-aways? How many experiments would we need to run to identify actionable take-aways, and would it scale?

**Potential Impact On The Field Of Automl:**

If ALBATOR becomes adopted by the AutoML community as the gold standard for evaluating models and their hyperparameters, it could have large impact within the field. However, it has yet to be proven in this regard, and the devil is always in the details. It is unclear what if any restrictions ABLATOR has that would limit adoption compared to using ray natively, nor why ABLATOR is necessary when Ray-Tune exists.

I see two main components to ABLATOR. The first is the actual system for running experiments, and the second is the result analysis. To me, the experiment running system is directly competing with Ray-Tune, and I don't see why one couldn't just use Ray-Tune to get results, and then analyze them with ABLATOR's analysis logic. (I suppose the main selling point is the experiment persistence?)

For this reason, I consider it to have medium potential impact, but if it manages to gain adoption and prove superior to Ray-Tune then the impact would massively increase.

**Review Confidence:**

3: You are fairly confident in your assessment. It is possible that you did not understand some parts of the submission or that you are unfamiliar with some pieces of related work.

**Review Rating:**

6: Borderline Leaning Accept: Technically sound submission where reasons to accept outweigh reasons to reject. Please use sparingly.

**Review Summary:**

I am borderline leaning accept on the condition that my concerns are addressed in the rebuttal, particularly regarding Table 1.

[Update]: While the rebuttal answered some of my more minor questions, the core questions on actionable take-aways & table 1 were not addressed sufficiently, and thus I am updating to borderline leaning reject, though this is a very close call overall as the paper has a lot of strengths.

[Update 2]: The authors have convinced me that the focus should generally lie on the utility of the tool rather than the technical rigor of the meta-analysis section, and thus I am updating my score to a 6 (borderline leaning accept).

**Technical Quality And Correctness:**

In general the approach and design of ABLATOR makes sense and is reasonable, and the experiments look to be reproducible.
My primary concern is the author's take-away in table 1 that seems to directly contradict the data.

## Table 1

Table 1: So FT-T is almost always better? What is the take-away here?
- "We find Tablator performs similarly in most datasets and outperforms in other." : This appears to be completely unsupported. FT-T wins on 10 of 11 datasets in Table 1, often dramatically (CA, HE, AL, CO). The only time Tablalor wins is on YE, and only by a small 0.9% relative error difference.

## TPE

- "When TPE was used, all optimizers appeared to underperform on average by 4.6% and 3.8% when evaluating the best and mean trial performance."
  - I don't understand what this means, nor what it is being compared to. "All optimizers" seems odd, wouldn't SGD have better certainty given it was sampled more often? Why would the mean trial performance underperform random search due to undersampling? Or is TPE underperforming random search in general as an HPO algorithm?

## Survivor Bias
- "We conclude that smaller models and dataset will have a Survival Bias from the fewer out-of-memory execution errors and that such bias could be mitigated by running separate experiments with sufficient resources allocated."
  - Technically correct, but "Just use a bigger machine until it doesn't fail" is a trivial conclusion that doesn't actually solve the problem when you can't use a bigger machine. I don't see how ABLATOR resolves the survival bias problem meaningfully besides inflicting constraints that demand all trials succeed when in reality this can be impossible.

---

### Official Review · Reviewer_cLxo · 2023-05-16

**Potential Impact On The Field Of Automl:** NA, Reproducibility Review.
**Potential Impact On The Field Of Automl Rating:** 3
**Technical Quality And Correctness:** N/A
**Technical Quality And Correctness Rating:** 3
**Clarity Rating:** 3

**Summary Of Contributions:**

N/A, Reproducibility Review.

**Actions Required To Increase Overall Recommendation:**

Writing a basic `demo.py` would greatly help the user understand how the code works.

**Clarity:**

Pros:
* Codebase has well-organized files alongside a concise README.
* Code is Pytyped everywhere, which is very great for readability and run-time build-checking.


Cons:
* It would be nice to add a `demo.py` for a user to check if they've properly installed the package, and how the package should be used.
* Small coding style issues, ex:
  * `__init__.py` files should generally be lightweight and only import from source files, not define entire classes: https://github.com/fostiropoulos/ablator/blob/abcd/ablator/modules/loggers/__init__.py


**Overall Review:**

N/A

**Reproducibility (Optional):**

1. I was able to `pip install` the library with no issues.
2. However, because there was *no immediate `demo.py`*, it wasn't clear exactly what to run from this library.
3. The next best thing I could do was check the pytests in this library, and here are the following runs and outputs (after `pip install pytest`):
  * `computer:~/ablator-abcd/tests/main$ python3 test_wrapper.py`
    * `Loss Diverged. Terminating. loss: inf`
  * `computer:~/ablator-abcd/tests/main$ python3 test_mp.py`
    * [Long Terminal Logs related to training]: `Finished training - e713_9991. Limit of trials to sample '10' reached. There are 6 complete trials. with ids: ['f30d_9991', '34c5_9991', 'd6f0_9991', 'af79_9991', '7198_9991', 'e713_9991']. There are 4 unfinished trials. with ids: ['f51f_9991', 'bd3d_9991', 'df1b_9991', '56ca_9991']`
  *  `computer:~/ablator-abcd/tests/main$ python3 test_trainer.py`
    *  `Metrics batch-limit 32 is smaller than the validation dataloader length 100.`
  * `computer:~/ablator-abcd/tests/main$ python3 test_state.py`
    * [No logs]

From the above, especially on `test_mp.py`, the code seems to be working (i.e. starts multiple work units, each with different sets of hyperparameters for evaluation).




**Review Confidence:**

4: You are confident in your assessment, but not absolutely certain. It is unlikely, but not impossible, that you did not understand some parts of the submission or that you are unfamiliar with some pieces of related work.

**Review Rating:**

8: Accept: Technically sound paper with major impact, with perhaps some minor flaws.

**Review Summary:**

N/A